# The Factors Affecting Frailty among the Elderly in Korea: A Study Using the Frailty Cohort

**DOI:** 10.3390/ijerph20010670

**Published:** 2022-12-30

**Authors:** DaSol Park, Hee-Sun Kim

**Affiliations:** 1Department of Occupational Therapy, Kyungnam University, Changwon 51767, Republic of Korea; 2Department of Health Policy Research, National Evidence-Based Healthcare Collaborating Agency, Seoul 04933, Republic of Korea

**Keywords:** activity of daily living, cognitive, elderly, frailty, mental, occupational therapy, physical, social

## Abstract

In this study, the effects of health-related factors such as cognitive, mental, and physical functions on frailty among the elderly in Korea were analyzed, and basic data were provided to prevent frailty. Based on the frailty cohort, this study classified 11 independent variables into physical, mental, cognitive, social, and activities of daily living. Multiple regression analysis was conducted using the Korean frailty index as the dependent variable. Consequently, the following regression equation was derived: The Korean frailty index = 8.412 + 0.148 × (SGDS-K) − 0.000095 × (K-ABC) − 0.122 × PF + 0.327 × (K-ADL) − 0.049 × (K-IADL) − 0.047(MNA) + e. Among the 11 factors, one mental (SGDS-K) and two activities of daily living were health-related factors affecting the frailty of the elderly. The results of this study will help in establishing plans for the prevention and intervention of aging and can provide basic data for government-level community care for the prevention of aging in the future.

## 1. Introduction

Korea’s population is rapidly aging, which is in line with the trend of a low birth rate, and the proportion of the elderly population is expected to gradually increase to 24.3% by 2030 and 40.1% by 2060 [1]. These changes increase the likelihood of diseases among the elderly who survive to old age and can be accompanied by physical, mental, and social disabilities from aging even if they do not develop diseases [2]. As the trend of aging accelerates around the world, interest in healthy old age is increasing, and aging is becoming an important topic [3].

Frailty has a lower physiological reserve capacity to maintain individual homeostasis and an overall decline in function, which increases the likelihood of developing multiple diseases, resulting in increased functional dependence or hospitalization [4]. In 2012, a joint meeting of six academic organizations around the world defined physical senescence as “a medical syndrome that increases individual vulnerability to dependence and death due to deterioration of physical strength, endurance, and physiological function due to various causes and triggers” [5]. The cycle of frailty depicts a vicious cycle in which energy consumption declines because of decreased appetite, resulting in low nutritional status, which not only reduces overall physical function but also decreases fatigue, vitality, and muscle strength, resulting in reduced activity and energy consumption [2]. In addition, as people age, irreversible disorders develop, resulting in increased hospitalization, difficulty in completing recovery, and a lower quality of life for elderly patients due to the need for nursing homes, ultimately increasing the mortality rate [2].

The prevalence of senescence in the aged population varies according to the definition of a measurement tool, although it has been reported that approximately 41.9% of pre-frailty and 10.7% of frailty occur outside the country [6].

According to other international surveys, frail adults are six times more likely to die within three years than healthy seniors, and 2.63 times more likely to progress to frailty within three years of entering the pre-frailty stage [7].

The prevalence of frailty is expected to be high in Korea because of rapid aging, with the degree of frailty in the elderly in Korea being 7.7% to 21.3% and the pre-frailty stage being 25.5% to 49.8% [8]. In addition, a comparison of the prevalence of frailty in Korean cities and rural areas revealed that rural areas had a higher proportion of elderly people than cities, indicating that the imbalance of aging between communities is also deepening [9].

It has been reported that the cause of frailty in the elderly is not a single factor, but various factors accumulate in combination to produce negative health results. According to studies reported abroad, frailty is commonly observed in older people, particularly older women, and it is also related to lower education and income [7,10]. It is also associated with daily life dysfunction and chronic diseases such as chronic kidney disease, depression, stroke, and respiratory disease [11]. In Korea, however, not many studies have been conducted to identify the factors associated with aging in the elderly. Apart from the studies of Jeong et al. (2013) and Kim et al. (2021), most studies either derive frailty results from a single factor or were derived from very little research [12,13].

In this study, the effects of health-related factors such as cognitive, mental, and physical functions on the aging of the elderly in Korea were analyzed, and basic data were provided to prevent frailty. The specific objectives of this study were as follows:

First, health-related factors (cognitive, mental, physical, and social functions, as well as daily life activities), excluding the general characteristics of the subjects, were identified. Second, we considered a preventive approach for frailty using the derived elements.

## 2. Materials and Methods

### 2.1. Collection of Research Subjects and Data

This study used the “Old Age Frailty Cohort Data” by the Korean Old Age Frailty Cohort Project, which recruited and followed up on 3011 elderly people aged between 70 and 84 living in local communities nationwide for two years from 2018 to 2019. Therefore, this study is a cross-sectional study. Data collection was conducted at ten cohort research centers nationwide for those who agreed to participate in the study. It included 96 pages of questions. In this study, approximately 80 questions from a 96-page questionnaire were selectively used based on the purposes of the research and theoretical background.

### 2.2. Independent Variables

Based on previous studies, the independent variable was selected as the factor affecting frailty. All the variables used were verified for reliability and validity, and only tools with a history of use in the clinical field were selected. According to the evaluation tool’s intended use at the time of development, the sections of the tool were classified into cognitive function, mental function, physical function, social function, and daily life activities.

#### 2.2.1. Cognitive Function

##### Korean Version of Mini-Mental State Examination (MMSE-K)

The Korean version of the Mini-Mental State Examination (MMSE-K) is a cognitive test comprising 30 questions in seven categories: time-lapse, space-lapse, memory registration, memory recall, attention-focused and calculation, language function, as well as understanding and judgment [14]. It is a test tool commonly used in clinical trials in the Korean community, and the test takes 5 to 10 min. A total score of 24 points or more was set as “definite normal”, 20 to 23 points as “suspected dementia”, and 19 points or less as the criteria for “definite dementia” [14]. Cronbach’s alpha value of MMSE-K was 0.85.

##### Frontal Assessment Battery (FAB)

The frontal assessment battery (FAB) is a test consisting of the following six sub-items indicating different functions associated with frontal lobe function. (1) Conceptualization (similarity test); (2) thinking flexibility (language fluency test); (3) exercise programming; (4) sensitivity to interference stimuli (conflict instruction); (5) inhibitory control. Each sub-item to the environment is scored from 0 to 3, with a total score of 18 [15]. Cronbach’s alpha value of FAB was 0.797.

##### Digit Span Test: Korean-Wechsler Adult Intelligence Scale (K-WAIS)

In this study, one of the 11 small K-WAIS tests, the number memorization test, which is one of the small tests included in the language test, was used [16]. This sub-examination measures auditory short-term memory and attention and consists of seven questions to memorize immediately and seven questions to memorize backwards. The total score is calculated by adding seven questions memorized immediately and seven questions memorized from bottom to top. The number memorization test is useful for evaluating elderly patients and those with dementia because the test instructions are simple and easy to perform. Cronbach’s alpha value of K-WAIS was 0.87.

#### 2.2.2. Mental Function

##### Korean Version of Short Geriatric Depression Scale (SGDS-K)

The Korean version of the Short Geriatric Depression Scale (SGDS-K) is a self-reported depression scale with 30 questions and is used to diagnose depression [17]. However, SGDS-K, a shortened form of GDS, was developed because of the disadvantages of the time required for GDS, therefore, in this study, SGDS-K, which was standardized in Korean, was used [18,19]. Each of the 15 questions was a simple yes/no question, with 1 point for a positive response and 0 points for a negative response. The total score ranges from 0 to 15 points, and the higher the score, the more severe the depression. Cronbach’s alpha value of SGDS-K was 0.89.

#### 2.2.3. Physical Function

##### Korean Version of Activities-Specific Balance Confidence Scale (K-ABC Scale)

The Korean version of the Activities Specific Balance Confidence Scale (K-ABC Scale) was used to measure balanced confidence [20,21]. Cronbach’s alpha value of K-ABC was 0.82. The scale consists of 16 specific activities and asks how confident one is to perform well without falling or losing balance in each activity. Participants responded within the range of 0% (not at all confident) to 100% (not completely confident). The average score obtained by dividing the 16 questions by the total number of questions (16) was considered the final score. A balanced confidence score of 80% or higher was interpreted as a high level of physical function, a score of 50–80% was interpreted as a medium level, and a score of less than 50% was considered a low level [22].

##### Strength, Assisted Walking, Rising from a Chair, Climbing Stairs, Falls (SARC-F)

To screen for sarcopenia, we used Kim’s tool, which is a Korean version of the SARC-F (Strength, Assistant Walking, Rising from Chair, Climbing Stairs, Falls) tool created by Malmstrom [23,24]. Cronbach’s alpha value of K-ABC was 0.7. This tool comprises five self-reported questions related to muscle strength, walking assistance, standing up from a chair, climbing stairs, and falling. The score per question ranged from 0 to 2 points, and a score of more than 4 out of 10 points was classified as sarcopenia. In this study, four questions were used except for the fall-related question “How many times have you fallen in the past year.

##### Physical Functioning (PF)

This evaluation tool was developed by Lee (2001), and its Cronbach’s alpha value was 0.92, indicating high reliability [25]. PF consists of a total of 10 questions, and five of those items are assigned to exercise ability and self-care. Regarding athletic abilities, three questions measured lower extremity exercise, and two questions measured upper extremity exercise ability. As the ADL and IADL evaluation tools were used separately in this study, only five questions about exercise ability were used. The PF scale was 1 (very difficult) to 3 points (not difficult at all) [26]. The larger the total score, the better the functional condition.

#### 2.2.4. Activities of Daily Living

##### Korean Activity of Daily Living (K-ADL or K-Basic ADL)

K-ADL is a tool developed by Katz et al. (1963), which was adapted to suit the situation in Korea by Won et al. (2002) and comprises a total of seven questions (dressing, washing, bathing, eating, moving, using toilets, and toilet control) [27,28]. Cronbach’s alpha value of K-ADL was 0.937.

##### Korean Instrumental Activity of Daily Living (K-IADL)

The K-IADL is a modification and supplementation of the IADL scale developed by Lawton and Brody (1969) to suit the living environment and culture of Koreans by Won et al. (2002) [28,29]. Cronbach’s alpha value of K-IADL was 0.935.

The K-IADL consists of seven 3-point scale questions (body grooming, housework, meal preparation, laundry, close-range outing, medication, and money management) and three 4-point scale questions (use of transportation, purchase of goods, and phone use).

##### Mini Nutritional Assessment (MNA)

The MNA is a tool developed based on body measurements and questions about diet and lifestyle to screen for malnutrition in the elderly [30]. The 18 questions in the MNA comprise four questions on body measurement (BMI, Mid-Arm Circumference: MAC, Calf Circumference: CC, Weight Loss), six general survey questions (lifestyle, drugs, physical activity, mental problems, ulcers, acute disease, two questions about health and nutritional status [31]. Cronbach’s alpha value of MNA was 0.83.

#### 2.2.5. Social Function

##### ENRICHD Social Support Instrument (SS)

ENRICHD (Enhancing Recovery in Coronary Heart Disease) consists of six questions, including emotional support, informational support, and instrumental support, with a yes or no response to each item [32]. A positive response to each item was assigned 1 point each, and a higher total score indicated better social support. The internal reliability of the tool was 0.93 for Cronbach’s alpha at the time of development.

### 2.3. Dependent Variable

#### Korean Frailty Index (KFI)

The Korean Frailty Index (KFI) is a multi-domain phenotype measurement tool consisting of eight items: hospitalization, subjective health status, drug use, nutritional status, emotional state, incontinence, walking ability, and communication disorder. Cronbach’s alpha value of KFI was 0.65.

### 2.4. Analysis Method

In this study, regression analysis was used to investigate health-related factors affecting the aging of the elderly in Korea. Multiple regression analysis was conducted as continuous variables with two or more independent variables and one dependent variable were used during the regression analysis. The purpose of multiple regression analysis is to create an equation for calculating the prediction *Y* from several independent variables. When *k* independent variables exist, the regression model equation is as follows:
*Y* = *a*_0_ + *b*_1_*X*_1_ + *b*_2_*X*_2_ ⋯ *b_k_X_k_*.(1)

*a_0_* is the regression constant (intercept *Y*) and is the regression coefficient of the independent variables. In this study, variables with significant F values were used as independent variables to derive the regression model equation. The regression analysis functions on the condition that there is no correlation between the independent variables in the basic assumption. Accordingly, the Durbin-Watson test and multicollinearity were verified. Frequency analysis was conducted for general information on the subjects using IBM SPSS Statistics 26 (Statistical Package for the Social Sciences, IBM Corporation, Armonk, NY, USA, 2019).

## 3. Results

### Results of Descriptive Statistics

(1)General information about the subject

Of the 3,014 subjects surveyed in the frailty cohort, 1407 (46.7%) were male and 1545 (51.3%) were female, excluding 62 with missing values (Table 1). The age of the survey subjects was distributed from a minimum of 72 to a maximum of 86 years, with an average of 78 (±3.88 years). Excluding 1734 missing values, 374 (12.4%) of the 1280 people graduated from elementary school, followed by No education (Illiteracy) with 249 (8.3%). The average number of years in education was 8.56 years. When asked, “Are you working on a job with an income?” Most of them stated they were retired, but 770 (25.5%) of the elderly continued to work.

(2)Analysis of aging influencing factors (multiple regression analysis)

Table 2 presents a summary of the model in which multiple regression analysis was conducted with frailty as the dependent variable. As the F change amount of the model is 118.5 and the significant probability F change amount is 0.000, it can be stated that the regression model is significant. In addition, the Durbin-Watson value was 1.878, located between 1.0 and 3.0, which is the criterion for evaluating the independence of a general residual, which has a value close to 2, and the significance probability was less than 0.001. Therefore, the regression model used in this study was determined to have no autocorrelation problem. The $R^{2}$ value of the model is 0.346; however, because there are many independent variables, it is appropriate to adopt the modified $R^{2}$ value. As the modified $R^{2}$ value was 0.343, it can be said that this regression model has an explanatory power of 34.3%.

The standardization coefficients and non-standardization coefficients, significance probabilities, and collinearity statistics of the independent and dependent variables are presented in Table 3. As the tolerance limit of all variables was 0.1 or more and the variance inflation factor (VIF) indicated values of less than 10, it was judged that there was no problem with collinearity. According to the model summary, the overall regression model was significant, but among the independent variables, the significance probability of MMSE, FAB, K-WAIS, SARC-F, and social support was higher than 0.05; therefore, it was not adopted in the regression equation.

Based on the non-standardization coefficients in Table 3, the regression equation for frailty is expressed as follows:

KFI = 8.412 + 0.148 × (SGDS-K) − 0.000095 × (K-ABC) − 0.122 × PF + 0.327 × (KADL) − 0.049 × (K-IADL) − 0.047 × (MNA) + e.

Some interpretations of the regression equation can be expressed as the higher the KFI by one unit, the higher the SGDS-K by 0.148, the lower the PF by 0.122, and the higher the K-ADL by 0.327. Examining the relative influence between the variables based on the standardization coefficient in Table 3, SGDS-K was 0.321 and PF was 0.228, which is the most influential variable for frailty. The facts that can be seen from the results of this study are as follows.

A high KFI score means that there is a high possibility of aging.The higher the depression (high SGDS-K score), the higher the possibility of frailty (KFI).The elderly (high K-ADL score) who do not perform basic daily life activities well have a high possibility of frailty (high KFI score).The worse the physical function (high PF score), the higher the possibility of frailty (high KFI score).The lower the balance confidence (high K-ABC score), the higher the possibility of frailty (high KFI score).Cognitive function (MMSE, K-WAIS, FAB) does not significantly affect the possibility of frailty (KFI)Social support (SS) does not significantly affect the possibility of frailty.

## 4. Discussion

This study attempted to analyze the effects of health-related factors, such as cognitive, mental, and physical functions, on frailty and provide basic data to prevent frailty. In the study, health-related factors affecting the frailty of the subjects were derived using frailty cohort data, and the results were analyzed to consider ways to prevent frailty. Among the 11 factors that had a statistically significant effect on frailty were one mental element (SGDS-K), two physical elements (K-ABC and PF), balanced confidence, physical exercise ability, and three daily life activity elements (ADL, IADL, and MNA).

The analysis of these results is as follows: First, cognitive function in the elderly did not have a significant effect on frailty. This result is consistent with the results of a study by Gobens et al. (2010) that found frailty in the elderly is not directly related to cognitive function [33]. The study by Gobens et al. (2010) is similar to this study in general demographic characteristics such as age, sex ratio, and education level of the subjects, and the MMSE was used in the same way in this study as a cognitive evaluation tool [33]. The Tilburg Frailty Indicator (TFI) is an evaluation tool for measuring frailty. As a result of the study, there was no direct correlation between MMSE and TFI, but a correlation between frailty and MMSE was found. Thus, it was observed that the MMSE indirectly affects TFI. In addition, a study by Borges et al. (2019) and Ma et al. (2019) reported that frailty leads to cognitive impairment [34,35]. Therefore, it is necessary to confirm the indirect correlation between cognitive function and age in future studies.

Second, social support for the elderly did not have a significant effect on frailty. The ENRICHD social support instrument used to confirm social support included emotional, informational, and instrumental support. Unlike this study, in the study by Gobens et al. (2010), the loneliness scale and social support list were used to grasp social support [33]. They set Frailty as a dependent variable, and social support, cognitive function, and physical function as independent variables. In addition, a correlation analysis between each variable was also conducted. The results of the study revealed that social support was not directly related to frailty but was correlated with the MMSE scores and physical function. This is partially consistent with the results of this study, suggesting the possibility that frailty can be indirectly reduced by improving social support in the future.

Third, based on the regression equation derived in this study, the factors that have the greatest influence on frailty in the elderly are depression as well as upper and lower extremity physical function. This is partially consistent with the results of Jang (2017) and Ribeiro et al. (2018) [36,37]. In addition, the results of this study were consistent with the study by Kim et al. (2021), who performed a multiple regression analysis on senescence, which concluded that the three most influential factors were depression, physical activity, and subjective health [13]. The reason depression was identified as the most influential factor was attributed to the question measuring KFI, which included an element to confirm the emotional state.

Using the above results and the frailty prevention method, it is possible to recommend a direction for the development of frailty prevention. Existing methods for preventing frailty can be divided into primary, secondary, and tertiary methods of prevention. Primary prevention includes risk factor management, health promotion, and function maintenance for non-frail elderly; secondary prevention includes early detection of signs of frailty and proper management for pre-frailty among the elderly; and tertiary prevention includes intensive management to restore the function of frail elderly and prevent the negative health consequences of frailty [38]. In addition, as primary prevention increases, community-centered treatment is provided, and as tertiary prevention increases, medical and nursing facility treatments are provided. Although multi-factor interventions such as exercise and nutritional therapy interventions are being performed according to each classification, specific prevention and intervention programs have not been developed [39].

The advantage of this study is that the reliability of the study is high because the data were derived using only standardized evaluation tools for 11 factors that are thought to affect frailty. In addition, among the 11 factors, the influential factor is a factor that predicts frailty and can be used as an intervention method for health promotion by improving the factor before frailty. Factors that do not affect can reduce unnecessary waste of economic and human resources by setting their importance low in hospitals or elderly care facilities. However, since the number of subjects in this study is small, it is insufficient to generalize only the results of this study. It is necessary to expand related research in the future.

Based on the results of this study, it is necessary to select high-priority tasks among the ADL tasks performed by the elderly in their daily lives to ensure that direct tasks can be performed to effectively prevent frailty. ADL tasks should be configured to improve mental and physical conditions, which are factors that have a significant influence on aging. It is necessary to focus on the primary preventive aspect because, in modern society, it is important not only for the elderly with diseases but also for the prevention of the overall frailty of the elderly living in the community. Therefore, recommending specific ADL tasks that can improve the psychological state and physical function of the elderly living in the community in terms of primary prevention could be a clue to preventing frailty. Proposing ADL tasks is within the scope of the occupational therapist’s job description, therefore, effective collaboration within the community is necessary.

The following are some of the limitations of this study: First, there is a limit to generalizing the effects of the 11 independent variables employed in this study because the KFI used as a dependent variable cannot represent the entire frail population Therefore, to apply the results of this study in clinical practice, it is essential to carefully confirm the detailed items of the KFI. Second, although cognitive and social support did not reveal statistically significant outcomes in the research findings, they could have an indirect correlation with physical function. In addition, it should be remembered that it can have a significant impact on people from other cultures and races.

## 5. Conclusions

This study sought to determine the effects of health-related factors such as cognitive, mental, physical, and social functions as well as daily life activities on frailty among the Korean elderly and to determine how to prevent frailty. Based on previous studies, 11 independent variables were classified into daily life activities as well as mental, physical, and cognitive factors, and a multiple regression analysis was conducted using the Korean frailty index as a dependent variable. As a result of the analysis, one mental element (SGDS-K), two physical elements (K-ABC and PF), balance confidence, physical exercise ability, and three daily life activities (BADL, IADL, and MNA) had a significant effect on frailty.

Among these, depression had the greatest impact, followed by physical factors and ADL. Based on the findings of the research, to effectively prevent frailty, it is necessary to select high-priority tasks among the ADL tasks performed by the elderly in their daily lives to ensure direct tasks can be performed. In modern society, the elderly aim to age with dignity. In other words, rather than ending one’s life in a hospital or nursing home, they aim to enjoy life as independently as possible in the same place they have lived. This study is significant because it provides basic data for community care policies for the elderly from the perspective of primary prevention.

## Figures and Tables

**Table 1 ijerph-20-00670-t001:** General information about subjects used as control variables.

		Frequency (*n*)	Percent (%)
Gender	Male	1407	46.7
	Female	1545	51.3
Age		78 (±3.88)	
Education	No education (Illiteracy)	54	1.8
	No education	249	8.3
	Elementary school	376	12.5
	Middle school	188	6.2
	High school	220	7.3
	College	41	1.4
	University	151	5.0

**Table 2 ijerph-20-00670-t002:** Regression Model Summary.

Changes in R Square	Changes in F	Probability of Significance F Variation	Durbin-Watson Value
0.346	118.517	0.000	1.878

**Table 3 ijerph-20-00670-t003:** Standardization coefficients (B)/non-standardization coefficients (β), significance probabilities, and collinearity statistics of independent and dependent variables.

	B	Error	β			Tolerance	VIF
MMSE	−0.021	0.011	−0.040	−1.788	0.074	0.543	1.843
FAB	−0.015	0.012	−0.028	−1.222	0.222	0.512	1.953
K-WAIS	−0.007	0.010	−0.016	−0.744	0.457	0.599	1.668
SGDS-K	0.148	0.009	0.321	16.114	0.000	0.669	1.495
K-ABC	−0.000095	0.000	−0.040	−2.294	0.022	0.885	1.131
SARC-F	0.000	0.034	0.000	0.010	0.992	0.281	3.560
PF	−0.122	0.017	−0.228	−7.326	0.000	0.273	3.662
K-ADL	0.327	0.064	0.090	5.110	0.000	0.849	1.177
K-IADL	−0.049	0.009	−0.094	−5.520	0.000	0.920	1.087
MNA	−0.047	0.011	−0.078	−4.190	0.000	0.755	1.324
SS	−0.002	0.004	−0.010	−0.590	0.555	0.947	1.056

## Data Availability

The data that support the findings of this study are available upon request from the corresponding author [H.-S.K.]. The data are not publicly available owing to restrictions (e.g., they contain information that could compromise the privacy of the research participants).

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
