# Peer review of "The Factors Affecting Frailty among the Elderly in Korea: A Study Using the Frailty Cohort"

_ijerph, 2022, doi:10.3390/ijerph20010670_

Round 1

Reviewer 1 Report

The authors point to an important issue regarding frailty syndrome in the growing elderly population and look for factors that are not indifferent to it.

The information contained in the introduction regarding the numerical representation of the stage before frailty and the frailty syndrom in the elderly population in Korea is completely ignored in the research section. With such a large population and KFI score, perhaps it would be appropriate to differentiate the respondents, this would undoubtedly increase the value of the analyses presented, and would also increase the consistency of the theoretical introduction with the empirical part.

The Cronbach's alpha coefficients of the used scales should be included. used scales should be included.

The description of the sample is incomprehensible (lines 203-204).

The figures presented in the text and in the table are not consistent.

The usefulness of the presented results for the development of frailty prevention is unclear and declarative. If the article is to add value, Authors should specifically illustrate which results in what context and how to use them to achieve the stated goal.

The results discussion section needs to be rethought and rewritten. Authors emphasize the importance of cultural differences, so in demonstrating the consistency of their own results with reports in the literature, they should describe in detail how the variables were operationalized, e.g., the ENRICH tool in not equal to Gobens et al. (2010) operationalizing loneliness and social support (line 263)

Authors should comment on the strengths and weaknesses of the entire study, not just parts of it.

I believe that this article suffers from some shortcomings that should and can be corrected after further consideration. I would be happy to see a revised version.

Author Response

Thank you very much for your reply. We wrote the correction in the attached file.

Reviewer 2 Report

Abstract

Please revise the sentence in the abstract 

  two activities of daily living were health-related factors affecting the aging of the elderly.

Do you mean the frailty by saying ageing ?

Method 

a 96-page questionnaire

How do you motivate older participants to answer such numerous questions? Readers can learn from you. Please describe the details.

Study design: Authors should state clearly that 

"this study is a longitudinal cohort study"

Please check the source of citation and add to the paper.

H. Jacobsen, Kathryn. Introduction to Health Research Methods: A Practical Guide (p. 74). Jones & Bartlett Learning, LLC. Kindle Edition. 

Objective;

Those are clearly written,

Results

Authors dis not report baseline and follow up level of indicators, Baseline level at the beginning the follow up and at the end of the follow up should be presented.

What is the length of follow up time?

Why authors did not choose longitudinal follow up, time-event analysis and cox regression if this is a cohort study?

Please provide  adequate reason. Otherwise the research will be cross-sectional data analysis.

If authors chose to a cross-sectional study, how authors decide dependent and independent variables in term of temporality?

The paper deserved a major revision  

Author Response

(The authors gave the same response as above.)

Round 2

Reviewer 2 Report

Revision is satisfactory.